# Have Bilateral Free Trade Agreements (BFTAs) been beneficial? Lessons learned from 11 U.S. BFTAs between 1992 and 2017

Tareg Ghazi Alghabbabsheh[1], Saleh Saud AlSaif[2], Md. Saiful Islam [2‡]*, Tareq Saeed AlShammari[3‡], Ali M. A. Mahmoud[4]

1 Department of Economics, Faculty of Economics and Administrative Sciences, The Hashemite University, Zarqa, Hashemite Kingdom of Jordan, 2 Department of Economics & Finance, College of Business Administration, University of Hail, Hail, Kingdom of Saudi Arabia, 3 Department of Law, College of Law, University of Hail, Hail, Kingdom of Saudi Arabia, 4 Department of Economics, School of Politics and Economics, Beni-Suef University, Beni-Suef, Egypt

☉ These authors contributed equally to this work.
‡ MSI and TSA also contributed equally to this work.
* saifecon@yahoo.com

**Data Availability Statement:** All relevant data are within the paper.

## Abstract

This study analyzes aggregate data on trade flows to examine the performance of bilateral free trade agreements (BFTAs) between the United States and 11 BFTA nations in a two-dimensional approach. In line with the literature, this study applies the gravity model and analyzes the effect of the treatment using Poisson pseudo-maximum likelihood (PPML) panel data from 1992 to 2017. We use the PPML as an alternative methodology to an ordinary least squares model, as it can treat zero trade values and lead to unbiased estimates and different consistencies. We consider the trade in goods but excluded services because of the different nature of trade for services. Moreover, this study highlights the quantitative performance of BFTAs without considering the industry level to compare the overall benefits for the trade flows from the U.S. as an exporter to BFTA countries and vice versa. It thus adds to the debate on the effect of FTAs on trade flows and conducts pre-FTA and post-FTA analyses to compare the volumes of exports and imports during both periods. Findings vary according to the direction of trade; notably, some trading partners increased their trading volume to 388%. In sum, this paper provides a collective current state assessment to demonstrate the most plausible reasons for the effects of the 11 BFTAs, in addition to informing policymakers on the lessons learned from each BFTA.

## Introduction

When it comes to the benefits of trade agreements, a country reducing its trade barriers, that is, a country that applies trade liberalization, gains most from trade, either economically or politically. Although multiple factors can explain the increase in the world trade in goods and services, free trade agreements (FTAs) are often discussed by both the proponents and opponents of free trade. Proponents' arguments are driven by the overall economic benefits of

**Funding:** This research has been funded by Scientific Research Deanship at the University of Ha'il, Saudi Arabia through a project numbered RG-21 145. The authors would like to thank the University of Ha'il for its financial support.

**Competing interests:** The authors have declared that no competing interests exist.

FTAs, which can lead to some improvements in international markets, whether from the perspective of access to international markets or from increased efficiencies in terms of costs the benefits for both producers and consumers [1]. Politically, FTAs are often considered a base for strengthening the relationships between two countries and stabilizing a particular region. Especially in the case of bilateral FTAs (BFTAs), many pro-trade policymakers consider BFTAs as a means to expand the status quo of smaller U.S. trade partners by granting access to the larger U.S. market and providing the member states with less restrictive international trade policies. In return, this leads to higher per capita growth than trading with other countries that maintain international trade restrictions. However, BFTA opponents continue to question the reason for the existence of some FTAs and doubt the real benefits to the U.S. economy. Others challenge the overall potential of trade liberalization and globalization. Such reservations have hindered trade negotiations in the past few years, making this topic more sensitive than ever [2]. As a result, traditional barriers to trade have emerged as policy responses, and tariff scheduling has become deadlocked due to the rise in protectionism. Additionally, an escalation in trade wars imposed a threat to the global trading system [3].

Conversely, several international organizations, such as the World Trade Organization (WTO), the World Bank Group, the International Monetary Fund, and non-governmental organizations such as the World Economic Forum (WEF) [4] have been promoting trade liberalization agreements and developing policy options for multiple challenges to support the positive impacts of FTAs on the global economic outlook. According to WEF, reducing supply chain barriers to trade would boost the world's GDP six times more than the elimination of all existing tariffs. The WTO also reported that, in the early 2000s, the total global trade was 20 times higher than that in the 1950s [5]. The WTO report stated that, in the preceding decade, the global goods trade increased by 4.3% as a direct trade-related benefit. Additionally, G20 member countries have also been focusing on trade issues that could lead to the most significant gains. Some of the proposals include services trade liberalization, which would halve the global current account imbalances and the political tensions these imbalances would fuel. Promoting trade liberalization in services would require integrating small and medium enterprises (SMEs) into the trading system to share the benefits of trade throughout communities and support digital trade and the benefits of supporting value chains and new sources of growth [6].

Therefore, this study aims to highlight the quantitative performance of BFTAs collectively and compare the benefits of trade flows for the case when the U.S. is an exporter to BFTA countries and vice versa. Overall, this study considers the U.S. perspective on global trade. Additionally, it only considers trade in goods and merchandise but excludes services from the calculations. In line with the literature, this study applies the gravity model and analyzes the treatment effect using Poisson pseudo-maximum likelihood (PPML) panel data from 1992 to 2017 on 155 countries. As robustness test, pre- and post-FTA tests have been conducted. As mentioned above, the study provides a collective analysis in a bi-directional manner for the United States side once as an exporter and as an importer. This approach will demonstrate the most plausible estimates of the average effects of a BFTA on the bilateral trade flow, which are obtained from a theoretically motivated gravity equation after applying country and time fixed effects.

The remainder of this paper is structured as follows. The next section presents the history of FTAs and provides the background of multilateral FTAS in addition to BFTAs. The subsequent section reviews the international trade policies and studies in the literature. The fourth section presents the gravity model, discusses the FTA dummy variables, and describes the data sources. The following section reports the results of bi-directional BFTA trade flows. Finally,

the last two sections discuss the findings from actual agreements and present policy implications and future research directions, respectively.

## Background

FTAs are viewed as crucial trade liberalization deals, as the negotiations are aimed at stimulating trade deals between member countries. Many FTAs are constructed bilaterally, that is, countries give each other preferential trade treatment, such as reducing tariffs on certain goods or other entry barriers. However, the goods must comply with all requirements and meet the compliance guidelines mentioned in the agreement. For example, in the case of the BFTA between the U.S. and Australia that entered into force in 2005, Australia reduced its tariffs on most U.S. agricultural and manufactured goods. In return, the United States decreased its tariffs on Australian beef, dairy, and other food items. The earliest forms of U.S. FTAs were implemented as BFTAs in 1985 with Israel and in 1989 with Canada. The growing benefits of BFTAs have led to increased interest in establishing even more BFTAs as interest has been explicitly expressed by several U.S. administrations in the past [7]. For example, the Bush and Obama administrations have both been interested in establishing BFTAs with more members of the Association of Southeast Asian Nations (ASEAN) and Middle Eastern countries.

There is also another type of FTA categorized as a multilateral agreement, also called a regional trade agreement [8]. For instance, the Central America Free Trade Agreement-Dominican Republic (CAFTA-DR) is mostly a group of bilateral deals between the United States and member countries within the Central American region.

As presented in Table 1, there are two MFTAs (NAFTA and CAFTA-DR) and 12 BFTAs in total. The U.S.–Canadian BFTA became part of the first U.S. MFTA, known as the North American Free Trade Agreement (NAFTA) [9]. This agreement was expanded to include

**Table 1. Years of establishment of all FTAs and trade members from 1992 to 2017.**

| Year | Country | Trade Agreement |
|------|---------|-----------------|
| 1985 | Israel | BFTA |
| 1994 | Canada | NAFTA |
| 1994 | Mexico | NAFTA |
| 2001 | Jordan | BFTA |
| 2004 | Chile | BFTA |
| 2004 | Singapore | BFTA |
| 2005 | Australia | BFTA |
| 2006 | Bahrain | BFTA |
| 2006 | El Salvador | CAFTA-DR |
| 2006 | Guatemala | CAFTA-DR |
| 2006 | Honduras | CAFTA-DR |
| 2006 | Morocco | BFTA |
| 2006 | Nicaragua | CAFTA-DR |
| 2007 | Dominican Republic | CAFTA-DR |
| 2009 | Costa Rica | CAFTA-DR |
| 2009 | Oman | BFTA |
| 2009 | Peru | BFTA |
| 2012 | Colombia | BFTA |
| 2012 | Panama | BFTA |
| 2012 | South Korea | BFTA |
| 2020 | Japan | BFTA |

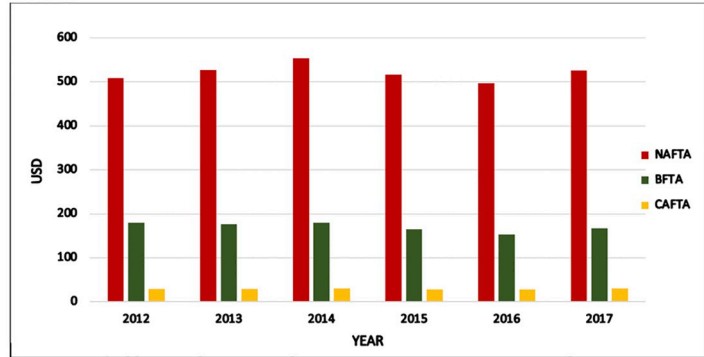

**Fig 1. Total exports from the United States to all FTAs (NAFTA, BFTA, and CAFTA-DR) from 2012 to 2017.**

Mexico, Canada, and the United States, and was established in 1994 [10]. The second MFTA was launched 12 years later, in 2006, incorporating six Central American countries: Costa Rica, El Salvador, Guatemala, Honduras, Nicaragua, and the Dominican Republic. The rest were all BFTAs focusing on developing countries, such as Jordan, Bahrain, Colombia, Oman, Peru, Panama, Chile, and Morocco, and three other BFTAs with developed economies, Singapore, Australia, and the Republic of Korea.

One of the most recent developments occurred in 2019: the BFTA negotiations between the United States and Japan. The U.S. provided a reduction in tariffs and a complete elimination of tariffs for some of the 241 tariff lines recognized by the U.S. Customs Border Protection. The policy design of this agreement primarily secured market access for agricultural goods and industrial merchandise. Consequently, the U.S.–Japan agreement was approved and issued in December 2019 and was fully implemented in January 2020. According to the Fact Sheets published by the Office of the United States Trade Representatives, more than 90% of the U.S. agricultural goods exported to Japan will also be duty-free or benefit from tariff reductions. Figs 1 and 2, present the magnitude of the trade-related benefits of such agreements captured by the share of the trade flows between exports and imports. For instance, in 2017, NAFTA accounted for more than 77% of trade, which is the largest share among all FTAs [11]. While CAFTA-DR accounted for a small portion of all trade flows, which was not more than 4% only. The rest of the trade shares were around 19%, distributed across all 12 BFTAs.

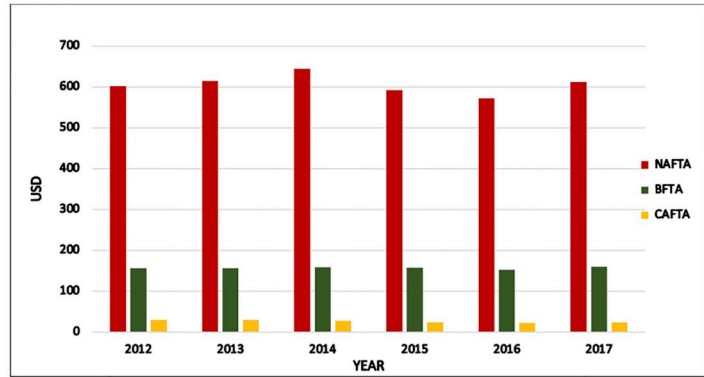

**Fig 2. Total imports to the United States from all FTAs (NAFTA, BFTA, and CAFTA-DR) from 2012 to 2017.**

Table 2. U.S. total trade for goods for 1992–2017 (seasonally adjusted).

| Year | Exports | Imports | Trade Balance |
|---|---|---|---|
| 1992 | 448,163.5 | 532,664.4 | -84,500.9 |
| 1993 | 465,091.0 | 580,659.0 | -115,568.0 |
| 1994 | 512,627.0 | 663,254.0 | -150,627.0 |
| 1995 | 584,742.0 | 743,543.0 | -158,801.0 |
| 1996 | 625,077.0 | 795,290.0 | -170,213.0 |
| 1997 | 689,181.0 | 869,703.0 | -180,522.0 |
| 1998 | 682,137.0 | 911,896.0 | -229,759.0 |
| 1999 | 695,798.0 | 1,024,618.0 | -328,820.0 |
| 2000 | 781,918.0 | 1,218,021.0 | -436,103.0 |
| 2001 | 729,100.0 | 1,140,999.0 | -411,899.0 |
| 2002 | 693,104.0 | 1,161,366.0 | -468,262.0 |
| 2003 | 724,771.0 | 1,257,121.0 | -532,350.0 |
| 2004 | 814,874.5 | 1,469,703.0 | -654,828.5 |
| 2005 | 901,081.8 | 1,673,456.0 | -772,374.2 |
| 2006 | 1,025,969.0 | 1,853,939.0 | -827,970.0 |
| 2007 | 1,148,197.0 | 1,956,962.0 | -808,765.0 |
| 2008 | 1,287,441.0 | 2,103,641.0 | -816,200.0 |
| 2009 | 1,056,042.0 | 1,559,625.0 | -503,583.0 |
| 2010 | 1,278,493.0 | 1,913,858.0 | -635,365.0 |
| 2011 | 1,482,507.0 | 2,207,954.0 | -725,447.0 |
| 2012 | 1,545,820.8 | 2,276,267.1 | -730,446.3 |
| 2013 | 1,578,516.9 | 2,267,986.7 | -689,469.9 |
| 2014 | 1,621,873.8 | 2,356,356.1 | -734,482.3 |
| 2015 | 1,503,328.4 | 2,248,811.4 | -745,483.0 |
| 2016 | 1,451,459.7 | 2,186,785.9 | -735,326.2 |
| 2017 | 1,547,195.4 | 2,339,591.3 | -792,395.9 |

**Note**: All figures are in millions of US dollars on a nominal basis and not seasonally adjusted unless otherwise specified. The individual figures may not equal the totals owing to rounding. The table reflects only the months for which there was trade.

Overall, the Pareto concept manifests itself in FTAs, with 80% of trade resulting from 20% of trade partners, namely the NAFTA members.

According to the U.S. Census Bureau, [12] the total trade in 2017 in goods between the U.S. and all countries amounted to USD 3.78 trillion. Imports were roughly USD 2.3 trillion, while exports were around USD 1.45 trillion, with a trade deficit of USD 792 billion or 21% (see Table 2). The FTAs accounted for approximately USD 1.5 trillion or 38% of the total trade in 2017. After breaking down the total into exports and imports, the amounts were around USD 770 billion for imports and USD 720 billion for exports, resulting in a total FTA trade balance of USD 50 billion, or 3.3%. Therefore, the FTA share of U.S. trade represents a somewhat more balanced trade than the trade deficits from trading with non-FTA nations.

## Literature review

Because a one-size-fits-all method does not account for the variety of factors that influence trade flows, the gravity model helps identify the factors associated with trade growth [13]. Another remodeled version of the gravity method, known as the "matching approach, " follows an experimental form with control groups. This approach involves the control groups of non-

FTA countries, yet with properties somewhat comparable to those of FTA countries, which allows to check for discrepancies between non-FTA and FTA members and only extract the relevant effects of an FTA [14]. Baier and Bergstrand adopted a complex model to capture the impact of FTAs on trade flows. They indicated that an FTA should not be realized as a dummy variable and also highlighted the paradox of causality and covariance. Further, they emphasized the importance of understanding what causes the creation of a certain BFTA and qualitatively investigated the background behind the establishment of such agreements. A question raised by Baier and Bergstrand in 2005 and 2009 is whether bilateral trade flows existed before or after the establishment of the BFTA [15, 16]. Therefore, two U.S. BFTAs were covered by Baier and Bergstrand (2005) [15]. They examined the cases of Israel (1985) and the BFTA with Canada (1989). They concluded that "an FTA will increase two member countries' trade by about 86 percent after 15 years" [15]. However, they believed that the 20 U.S. FTAs probably encountered varying results compared to the other 50 global FTAs examined in their 2005 paper.

Caporale et al. (2009) investigated the FTAs between the European Union and former Eastern European nations before EU integration. Using the gravity model, they affirmed that FTAs had a positive and significant effect on trade flows [17]. Sung Eun Jung (2012) considered a diversified list of international FTAs that mostly did not include the U.S., and his results showed a simple effect, which was positive but not statistically significant [18]. Frankel, Stein, and Wei (1995) did not detect any increase in the trade from European and EFTA countries and questioned the value creation of the established FTAs between the U.S. and the BFTA countries [19].

As mentioned in the previous sections, this study focuses on 11 bilateral FTAs. The application of a particular methodology to estimate the benefits of United States' bilateral trade with its 11 BFTA partners has always been accompanied by challenges in reflecting the unobserved variables that may be a source of improvement or deterioration for trade flows. Although researchers have examined trade creation extensively and attempted to broadly capture the effects of trade agreements, the results have been overall conflicting. Some have found that, on average, a country engaging in a new regional agreement is more likely to become a trade creator, with a future reduction in tariffs [20]. Nonetheless, in the late 1990s, the substantial empirical international trade literature tried to explain why trade agreements may not improve trade flows. The positive effects of FTAs on the overall global trade by member countries could be compensated for by other circumstances, such as a reduction in the trade with non-FTA members. Examining the direct link between FTAs and the bilateral trade capacity of trading nations is challenging for several reasons [21].

Leung (2016) investigated the effects of FTAs on bilateral vertical specialization (VS). They created a bilateral VS variable in the manufacturing between the U.S. and its trading members using input–output analysis, and then modeled an augmented gravity equation [22]. The results revealed that North American countries have the most significant results. The average treatment effect of an FTA is 0.94, which indicates that bilateral trade flows increases on average by 155% due to engagement in an FTA.

Overall, the observed models were not able to account for all influential trade factors. In the case of a biased upward estimate, the positive correlation between FTAs and trade dimensions could be weaker, and vice versa. Another justification stems from the existing causality between FTAs and trade capacity, which could be driving both. The higher the trade capacity is, the greater is the likelihood of creating FTAs, and this reverse causality produces an inaccurate representation of the relationship between FTAs and trade. Additionally, there could be measurement errors in the variables used in the empirical examination. Assume that the establishment of FTAs is significantly associated with members' political factors or economic

policies, which are further associated with trade dimensions. In this case, these other determinants must be considered to correctly understand the association between FTAs and trade volume. For instance, the empirical literature has revealed that EC members' size could define the increase in trade between EC countries in the 1960s and 1970s, as well as the infrastructure level, contiguity, shared borders, and common languages [23]. Similarly, the fast growth of East Asian economies is adequate to justify the rise in their trade share from 1965 to 1990 [24]. By contrast, Baier and Bergstrand (2007) explained that, even after considering most of their influence on trade flows, FTAs were still affirmed to enhance member countries' trade by approximately 86% in the following 15 years of being an FTA member [21].

Another significant hurdle in settling the dispute regarding the impact of FTAs on trade flows is the absence of consensus among scholars in using the gravity model. Although most international trade economists consider the gravity model in their investigation, the determining variables of bilateral trade flows are still debated. In general, the gravity equation shows a positive relationship between bilateral trade flows and the economic size of trading nations relative to the world economy, assuming no transaction costs [25]. An important explanatory variable is the distance between trading countries, seldom noted as the iceberg cost, as it is negatively associated with the trade volume between trading countries. Despite the lack of consensus in the literature on which other variables should be incorporated in the extended gravity model, some scholars assumed it might guide the selective reporting of results driven by researchers' previous beliefs [26].

Although it has been the norm in the literature to use the gravity model in the estimation of cross-sectional data, panel data on trade has also recently been used to analyze the effects of FTAs on trade flows. The implications appear to be higher and more significant when applying panel data compared to cross-sectional data. Researchers have demonstrated that panel estimates exhibit more credible and uniform results than cross-sectional estimates [20, 21]. However, few studies have analyzed the relationship between FTAs and trade volume for services compared with goods trade. Research on the service sector is lagging because of the comparatively later insertion of services into FTAs and the shortage of quantitative data on trade in services. Empirical examinations of service trade have used the gravity model to reveal the determinants of bilateral trade flows in services. Some studies have used a binary variable that symbolizes whether trading countries are in a multilateral trading agreement in the model, which is used more as a control than a policy variable of interest. Although the importance of the impact of FTAs on trade volume may vary across studies, Hur and Park (2012) found that the trade-creating results of FTAs are sector-specific, especially in the service sector [27]. In terms of the relationship between transportation costs (i.e., geographic variables) and trade agreements, the variables have been previously examined in the literature, beginning with Viner (1950), who considered the principle of the most favored nation (MFN) within Europe, which dates back to the nineteenth century [28]. Viner has reasoned them as the "close ties of sentiment and interest arising out of ethnological, cultural, or historical political affiliations." Other studies, such as Meade (1955) and Lipsey (1957), have not discussed the plausible influence of transportation costs on trade agreements [29, 30]. Wonnacott and Wonnocott (1981) attempted to determine whether the closeness between nations made FTAs more advantageous than non-FTAs [31]. In their study, they specified an essential of transportation costs and how it impacts FTAs, but their findings were criticized by many scholars since transportation costs have to be extremely high in order to hinder the performance of the FTA being examined. Frankel, Stein, and Wei (1995), however, [19] have confirmed the negative impact of transportation costs on trade agreements, which supports the findings of Wonnacott and Wonnacott (1981). Almost the majority of the studies indicate that transportation costs are exogenous, they are not distinctive from other costs, such as cultural ties or networks, but

transportation costs shape a critical point that still significantly impacts the formation of
FTAs.

## Methodology and data

### Methodology

The methodological approach of this study is based on the gravity model that considers the
analogy with Newton's universal law of gravitation to describe the patterns of bilateral aggre-
gate trade flows between the two countries. To do so, we need to substitute the mass of two
objects by the gross domestic product (GDP) of two countries and the distance by the square
of the distance between the countries. The starting point of the gravity equation is formulated
as follows:

$$F = A \frac{Y_i Y_j}{d},  \tag{1}$$

where F represents the amount of trade (imports, exports, or their sum) between the two coun-
tries; $Y_i$ and $Y_j$ represent the GDPs of country I and j, respectively; d represents the distance
between them; and A is a constant.

The application of the gravity model was undertaken by Anderson and Van Wincoop
(2003) [32] where they developed their gravity model, which assumed symmetric trade costs:

$$FX_{ijt} = \frac{P_{it} * P_{jt}}{PW_t} \left( \frac{D_{ij}}{R_{it} * R_{it}} \right)^{1-\theta},  \tag{2}$$

where $FX_{ijt}$ represents the bilateral exports from country i to country j at time t, $P_{it}$ is the index
of the attributes of exporter i in a specific year t, $P_{jt}$ denotes all importer-specific factor demand
in year t, $PW_t$ is the world output in period t, $D_{ij}$ reflects bilateral trade costs, $R_{it}$ and $R_{it}$ are
external and internal multilateral resistance variables, and $\theta$ denotes the elasticity of substitu-
tion. Alternatively, Eq (2) can be rewritten as after taking the natural log:

$$\ln FX_{ijt} = \ln P_{it} + \ln P_{jt} - \ln PW_t + (1-\theta)\ln D_{ij} - (1-\theta)\ln R_{it} - (1-\theta)\ln R_{jt} + \epsilon_{ijt}.  \tag{3}$$

By modifying Eq (3) to include the FTA effect, Baier and Bergstrand (2007) [21] used panel
data to estimate the gravity model as follows:

$$ln\, FX_{ijt} = ln\, P_{it} + ln\, P_{jt} - ln\, PW_t + (1-\theta)ln\, D_{ij} - (1-\theta)ln\, R_{it} - (1-\theta)ln\, R_{jt} +$$
$$FTA_{ij} + Comm\, language_{ij} + Common\, border_{ij} + ln\, \epsilon_{ijt}.  \tag{4}$$

In line with the literature on the determinants of the gravity model [21], this study employs
five types of variables: (1) macroeconomic indicators, which include the real GDPs of export-
ers and importers; (2) geographic characteristics, which incorporate the weighted distance
between countries i and j based on the most populated cities and the area exporters and
importers. Landlocked exporters and importers are represented by dummy variables taking 1
if an exporter or importer is landlocked, and 0 otherwise. Finally, common borders between
countries are also represented by a dummy variable taking 1 if the country pair has a common
border, and 0 otherwise. (3) The FTAs between the USA and other countries represent the
trade facilitation variables, expressed a dummy variable that takes 1 if the country has and FTA
with the USA, and 0 otherwise. (4) The set of cultural variables designed to capture common
cultural characteristics of a country pair (e.g., common official language) is a dummy variable
that takes 1 if the two countries share the same language; there is another variables of coloniza-
tion *Colony*, which is a dummy variable that takes a value of one if colonized and zero

otherwise. (5) Finally, the multilateral trade resistance (outward and inward) represents the remoteness between exporters and importers. The first equation representing the outward multilateral trade resistance for the exports from country i to country j depends on the trade costs across all markets, as shown in Eq (4).

$$Remoteness_{it} = \sum_j \frac{Distance_{ij}}{GDP_{jt}/GDP_{wt}} \tag{5}$$

$$Remoteness_{jt} = \sum_j \frac{Distance_{ij}}{GDP_{it}/GDP_{wt}}. \tag{6}$$

Eq (5) considers inward multilateral resistance, which accounts for the imports into country i from country j on trade costs across all possible suppliers. Formally, following Baier and Bergstrand (2007) this study applies country and time fixed effects [21]. Therefore, Eq (4) can be rewritten as:

$$ln\,FX_{ijt} = \delta + \phi_i + \eta_j + \varphi_{it} + ln\,P_{it} + ln\,P_{jt} - (1-\theta)ln\,D_{ij} + BFTA_{ij} + Landlocked_i$$
$$+Landlocked_j + Colony + Common\,Language_{ij} + Common\,Border + \upsilon_{ijt}. \tag{7}$$

$\delta$ captures world output multiplied by a constant term across all exporters and importers and $\phi_i$ represents exporter fixed effects, being dummy variables equal to 1 for a particular exporter. This approach also applies to importers. Term $\eta_j$ in panel data accounts for unobserved heterogeneity, which is constant for a given exporter across all importers and for a given importer across all exporters. Term $\varphi_{it}$ represents the year fixed effects for each country and takes 1 for a specific year and after the agreements, and zero otherwise.

$$\delta = -lnPW_t, \tag{8}$$

$$\phi_i = -(1-\theta)ln\,R_{it}, \tag{9}$$

$$\eta_j = -(1-\theta). \tag{10}$$

The error terms $ln\,\epsilon_{ijt}$ depends on higher moments and variance. Then, the expected value of the error term connects to more explanatory variables. Many researchers have suggested using the PPML as an alternative to the ordinary least squares (OLS) model, as the former can treat zero trade values and lead to unbiased estimates and further consistencies. Therefore, if the model is estimated using OLS, the estimator is biased and inconsistent. In addition, the PPML has advanced properties, such as using the numbers of fixed effects as dummy variables and including observations for which the trade value is zero. In any case, it is not necessary to use a dependent variable that follows a Poisson distribution. Moreover, as these properties are specific to the PPML, we can rewrite Eq (7) as:

$$FX_{ij} = exp[\delta + \phi_i + \eta_j + \varphi_{it} + lnP_{it} + lnP_{jt} - (1-\theta)lnD_{ijt} + BFTA_{ij} + Landlocked_i$$
$$+Landlocked_j + Colony + Common\,Language_{ij} + Common\,Border_{ij} + \upsilon_{ijt}. \tag{11}$$

Finally, the results on the PPML coefficients are more accurate for analysis than the OLS coefficients. Intelligibly, the dependent variable is expressed as flows rather than a logarithmic value. Inversely, the independent variables are logarithms of values interpreted as elasticities.

In general, several factors determine the quantitative consequences of FTAs. However, we restricted the analysis to trace the growth in bilateral trade volume from the importing side, as well as from the exporting side. There is a large body of literature on the gravity theory and its

various formulations. The economic and cultural dimensions and distance to trading allies are deemed to reveal trade flows. The gravity method is used with other regression models such as the PPML and OLS to explain the motives behind trading with a country [33]. Nearly all gravity models incorporate explanatory variables, such as social, cultural, economic, and historical measures of distance [34, 35]. For instance, Anderson and van Wincoop (2003) studied the critical implications of relative trade restrictions that determine trade between regions [32]. Anderson and Wincoop stated:""Trade between two areas depends on the bilateral barrier between them relative to average trade barriers that both regions face with all their trading partner"" [32]. Although this study is a non-parametric analysis, it is safe to assume that the investigation should detect an FTA"s success through its trade flows rather than some comparable trade flows between non-FTA countries. The main reason for this assumption is that some BFTAs have fewer years of observations, as in the case of the FTAs with South Korea, Colombia, and Panama, which were launched in 2012.

The potential bias in the estimates was managed by controlling for country fixed effects and country and time fixed effects. Simultaneously, a before-and-after FTA test was conducted as a robustness check. The logs of the variables were applied to ensure consistency with the basic gravity equation. To eliminate reverse causality, as higher trade may induce more FTAs to develop, our model is sensible to detecting the relationship between FTAs and trade in goods.

## Data

This study investigates the trade flow effect of BFTAs between the United States and 11 different countries using panel data from 1992 to 2017 for 155. However, a country that engaged in a BFTA before 1992 was excluded from the study because it was outside the study period and few observations could be obtained before 1985. Therefore, the data were drawn from multiple sources. At the macroeconomic level, exports and imports were obtained from the""*U.S. Trade in Goods by Country*"" database, publicly available through the U.S. *Department of Commerc"s Census Bureau* [36]. The real GDPs for exporters and importers were obtained from the *Penn World Tables* (PWT), downloaded from the Groningen Growth and Development Centre. The other indicators, that is, distance, common language, landlocked importers and exporters, common border, and colony were retrieved from *Centre d'Études Prospectives d'Informations Internationales* (CEPII) Gravity Database.

Table 3 reports the descriptive statistics for the total number of observations (8,060). The table shows the variables used in the empirical analysis, including the dummy variables, mean, standard deviation, and minimum and maximum values. The average trade flows of US exports and importers are shown in the table. The average for the United States as an exporter is around USD 6,084,424,763, with a maximum of USD 313 billion and a minimum of USD 100,000. These trade flows are relatively lower than those of the USA as an importer, with an average of USD 9,395,555,435, a maximum of USD 505 billion, and a minimum of USD 82,442. Another value that stands out in this table is the average log-distance between exporters and importers, 9.057, with a standard deviation of 0.455. Furthermore, the log real GDP for exporters and importers has an average of 13.66, with standard deviations of 3.142. The rest of the variables shown in Table 3 are dummy variables.

## Results

### The case of U.S. exports

Table 4 displays the before and after FTA analysis to examine the impact of FTAs on U.S. exports. It indicates the FTA trading partner, along with the pre-FTA and post-FTA periods, the corresponding total export values, and average export values in both periods. The averages

**Table 3. Descriptive statistics.**

| Variable | Description of variables | Mean | S.D. | Min. | Max. |
|---|---|---|---|---|---|
| $FX_{ij}$ | Trade flows | 7,739,990,252 | 3.0529 e (10) | 82,442 | 5.05 e (11) |
| | USA exports | 6,084,424,763.1 | 2.2834 e (10) | 100,000 | 3.13 e (11) |
| | USA imports | 9,395,555,434.8 | 3.6571 e (10) | 82,442 | 5.05 e (11) |
| $\ln D_{ijt}$ | Log distance between exporters and importers | 9.057 | 0.455 | 7.639 | 9.70 |
| $\ln P_{it}$ | Log real GDP of exporter | 13.66 | 3.142 | 5.221 | 16.72 |
| $\ln P_{jt}$ | Log real GDP of importer | 13.66 | 3.142 | 5.221 | 16.72 |
| $BETA_{ij}$ | BFTA | 0.035 | 0.185 | 0 | 1 |
| Common Language$_{ij}$ | Common language (dummy variable taking 1 if there is common language between the exporter and importer, and 0 otherwise) | 0.277 | 0.447 | 0 | 1 |
| Colony | Colony (dummy variable equal to 1 if the exporter/importer has been colonized, and 0 otherwise) | 0.019 | 0.137 | 0 | 1 |
| Landlocked$_i$ | Landlocked exporter (dummy variable taking 1 if an exporter is landlocked, and 0 otherwise) | 0.1 | 0.300 | 0 | 1 |
| Landlocked$_j$ | Landlocked importer (dummy variable taking 1 if an importer is landlocked, and 0 otherwise) | 0.1 | 0.300 | 0 | 1 |
| Common Border$_{ij}$ | Common border ij (dummy variable taking 1 if there is common border between exporter and importer, and 0 otherwise) | 0.0129 | 0.112 | 0 | 1 |

Note: The number of observations is 8,060.

U.S. exports saw a tremendous increase in all post-FTA period. Compared to the pre-FTA values, the benefits of the FTA can be clearly observed for U.S. exports. The highest average among all post-FTAs can be observed in the case of Oman, with average exports in the pre-FTA period of USD 462,154,279 and average imports in the post-FTA period of USD 1,148,952,317; this accounts for an increase of 248% in the flows of U.S. exports to Oman. Then comes Peru, with an increase in the U.S. exports to the country by 236%, followed by Colombia with an increase of 227%, from USD 7,778,488,099 in the pre-FTA period to around USD 17,665,504,958 in the post-FTA period.

Table 5 displays the PPML estimation results for the United State" case as an exporter to BFTA member countries without applying country-pairs and year fixed effects. The robustness

**Table 4. Bilateral U.S export to its trade partner before and after FTAs entered into force.**

| FTA partner | Before | After | Total exports | | Average exports | |
|---|---|---|---|---|---|---|
| | | | Pre-FTA | Post-FTA | Pre-FTA | Post-FTA |
| Australia | 1992–2004 | 2005–2017 | 66,002,063,393 | 121,566,661,842 | 5,077,081,799 | 9,351,281,680 |
| Bahrain | 1992–2005 | 2006–2017 | 3,424,713,240 | 8,166,374,332 | 244,622,374.3 | 680,531,194.3 |
| Chile | 1992–2003 | 2004–2017 | 30,816,838,117 | 117,554,056,017 | 2,568,069,843 | 8,396,718,287 |
| Colombia | 1992–2011 | 2012–2017 | 155,569,761,981 | 105,993,029,750 | 7,778,488,099 | 17,665,504,958 |
| Jordan | 1992–2001 | 2002–2017 | 495,000,000 | 18,780,725,619 | 49,500,000 | 1,173,795,351 |
| Morocco | 1992–2005 | 2006–2017 | 4,680,239,488 | 10,332,519,640 | 334,302,820.6 | 861,043,303.3 |
| Oman | 1992–2008 | 2009–2017 | 7,856,622,757 | 10,340,570,852 | 462,154,279.8 | 1,148,952,317 |
| Panama | 1992–2012 | 2013–2017 | 7,134,062,818 | 2,141,198,228 | 339,717,277 | 428,239,645.6 |
| Peru | 1992–2008 | 2009–2017 | 44,276,876,471 | 55,286,837,337 | 2,604,522,145 | 6,142,981,926 |
| Singapore | 1992–2003 | 2004–2017 | 199,112,046,740 | 244,823,469,190 | 16,592,670,562 | 17,487,390,656 |
| South Korea | 1992–2011 | 2012–2017 | 702,988,678,290 | 404,024,778,360 | 35,149,433,915 | 67,337,463,060 |

**Table 5. PPML estimation results without fixed effects for the case of the United States as an exporter.**

| | (1) | (2) | USA-AUS | USA-BHR | USA-CHL | USA-COL | USA-JOR | USA-MOR | USA-OMN | USA-PAN | USA-PER | USA-SGN | USA-KOR |
|---|---|---|---|---|---|---|---|---|---|---|---|---|---|
| $\ln D_{ijt}$ | -1.461 (0.0150) *** | -0.755 (0.035) *** | -0.774 (0.035) *** | -0.756 (0.035) *** | -0.776 (0.035) *** | -0.756 (0.035) *** | -0.756 (0.035) *** | -0.755 (0.035) *** | -0.775 (0.035) *** | -0.747 (0.035) *** | -0.754 (0.035) *** | -0.808 (0.035) *** | -0.760 (0.035) *** |
| $\ln P_{it}$ | 0.963 (0.007) *** | 0.888 (0.007) *** | 0.889 (0.007) *** | 0.888 (0.007) *** | 0.889 (0.007) *** | 0.888 (0.007) *** | 0.888 (0.007) *** | 0.888 (0.007) *** | 0.888 (0.007) *** | 0.888 (0.007) *** | 0.888 (0.007) *** | 0.893 (0.007) *** | 0.888 (0.007) *** |
| $\ln P_{jt}$ | 1.061 (0.008) *** | 0.987 (0.008) *** | 0.990 (0.008) *** | 0.987 (0.008) *** | 0.990 (0.008) *** | 0.987 (0.008) *** | 0.987 (0.008) *** | 0.986 (0.008) *** | 0.987 (0.008) *** | 0.988 (0.008) *** | 0.987 (0.008) *** | 0.999 (0.008) *** | 0.988 (0.008) *** |
| Common language ij | | 0.239 (0.019) *** | 0.230 (0.019) *** | 0.239 (0.019) *** | 0.241 (0.019) *** | 0.239 (0.019) *** | 0.239 (0.019) *** | 0.239 (0.019) *** | 0.239 (0.019) *** | 0.240 (0.019) *** | 0.239 (0.019) *** | 0.219 (0.019) *** | 0.241 (0.019) *** |
| Colony | | 0.148 (0.104) | 0.167 (0.105) | 0.148 (0.105) | 0.150 (0.104) | 0.148 (0.104) | 0.149 (0.105) | 0.147 (0.104) | 0.148 (0.104) | 0.146 (0.104) | 0.148 (0.104) | 0.199 (0.103) * | 0.151 (0.105) |
| Landlocked i | | -0.954 (0.104) *** | -0.951 (0.105) *** | -0.954 (0.104) *** | -0.952 (0.104) *** | -0.954 (0.104) *** | -0.954 (0.104) *** | -0.955 (0.104) *** | -0.954 (0.104) *** | -0.954 (0.104) *** | -0.954 (0.104) *** | -0.941 (0.103) *** | -0.954 (0.104) *** |
| Landlocked j | | -1.017 (0.136) *** | -1.004 (0.136) *** | -1.016 (0.136) *** | -1.006 (0.136) *** | -1.017 (0.136) *** | -1.016 (0.136) *** | -1.020 (0.136) *** | -1.017 (0.136) *** | -1.011 (0.135) *** | -1.016 (0.136) *** | -0.972 (0.134) *** | -1.010 (0.136) *** |
| Common border ij | | 1.003 (0.0530) *** | 0.984 (0.053) *** | 1.003 (0.053) *** | 1.004 (0.053) *** | 1.002 (0.053) *** | 1.002 (0.053) *** | 1.002 (0.053) *** | 1.003 (0.053) *** | 1.015 (0.053) *** | 1.005 (0.053) *** | 0.945 (0.052) *** | 0.999 (0.053) *** |
| BFTA ij | | | 0.600 (0.125) *** | 0.376 (0.626) | 0.812 (0.164) *** | -0.032 (0.215) | 0.451 (0.480) | -0.731 (0.444) * | -0.026 (0.541) | 1.334 (0.283) *** | 0.174 (0.244) | 1.854 (0.130) *** | 0.598 (0.130) *** |
| Observations | 8,060 | 8,060 | 8,060 | 8,060 | 8,060 | 8,060 | 8,060 | 8,060 | 8,060 | 8,060 | 8,060 | 8,060 | 8,060 |
| Exporter FE | NO | NO | NO | NO | NO | NO | NO | NO | NO | NO | NO | NO | NO |
| Importer FE | NO | NO | NO | NO | NO | NO | NO | NO | NO | NO | NO | NO | NO |
| Year FE | NO | NO | NO | NO | NO | NO | NO | NO | NO | NO | NO | NO | NO |

Note: Significant. codes: 0.01 '***' 0.05 '**' 0.1 '*'. Robust standard errors are between parentheses.

Model 1 is the reduced model with only distance, GDP for exporters, and GDP for importers. Model 2 includes cultural and geographic variables that are all dummy variables. No fixed effects (country-pairs and year-fixed effects) were applied.

checking tool can be seen in the results of models (1) and (2), where a negative sign with a high level of significance is observed in the case of distance and a positive sign with a high level of significance is shown for both the exporter's and importer''s GDP. Many researchers have found that distance, commonly accepted as a proxy for transaction costs, plays an essential role in trade demonstrated that the closeness between producers and consumers might also be articulated in trade-flow analysis.

Moreover, in the first column of Table 5, model (1) is the reduced form of the equation and only includes distance and the GDPs of exporters and importers at time t. A natural log was taken for the distance and both GDPs. In the second column, model (2) includes cultural dimensions, which are mainly dummy variables that take 1 for a common language, colonization, or a common border between the exporter and importer, and 0 otherwise. The common language between trading countries, which is a variable in the extended form of the model, is positively related to exports. The results match those of previous works, which determined a significantly positive association between a shared language and trade volume [21, 37]. This confirms the belief that commonalities lead to lower transaction costs among trading nations, leading to more bilateral trade.

The colonization variable (*Colony*) has also been used as a proxy for the""hysteresis effect"" in trade, as it shows the historical relationship between countries and how it impacts trading experiences. This is important because one-time modifications in the course of trading history could lead to lasting consequences, as stated by Eichengreen and Erwin (1996) [13]. It is logical to consider that nations with a common colonization history might favor collaborating and engage in more trade. However, this variable has not been statistically significant in previous studies or in our study. In the case of common borders, a common border variable is essential to represent the value of transportation costs. Another dummy variable in the extended model captures the geographic location of a landlocked trading partner. This geographical variable is expressed in two ways: landlocked i and landlocked j and represents whether country i or country j are easily accessible and have ports to facilitate trade. The variable has a high level of statistical significance with a negative coefficient, showing a negative relationship between the volume of exports and being a landlocked trading partner.

The following columns include the complete form of the equation with the addition of BFTA for each trading case, capturing the trade flow from the 11 BFTAs. The coefficients on all explanatory variables are presented with their statistically significances. A total of, 8060 observations were employed in the PPML regression and the explanatory variables revealed country-pair analyses. The results in Table 5 show that, for distance, which primarily symbolizes trade costs, all 11 BFTA countries exhibited statistical significance and a negative relationship between the volume of exports and distance. Therefore, the findings in Table 5 are in line with the literature, stating that distance is significantly and negatively related to trade.

Furthermore, country pair year fixed effects are applied in Table 6 to address endogeneity concerns. This table captures time-variant variables (GDPs and BFTAs), while time-invariant variables (geographic and cultural variables) are excluded. Both GDPs remain at a statistically significant level and have positive coefficients in terms of exports. Ultimately, the dummy variable shows statistical significance for seven BFTA countries, except for Australia, Jordan, Oman, and Korea. The remaining BFTA countries display statistical significance and mostly positive coefficients, except for Bahrain (-0.583) and Singapore (-0.781).

## The case of U.S. imports

Similarly, Table 7 displays the before and after analyses of the FTAs to examine their impact on US imports. Specifically, it shows the corresponding total import and average import values

**Table 6. PPML estimation results with country and time fixed effects for the case of the United States as an exporter.**

| | USA-AUS | USA-BHR | USA-CHL | USA- COL | USA-JOR | USA-MOR | USA-OMN | USA-PAN | USA-PER | USA-SGN | USA-KOR |
|---|---|---|---|---|---|---|---|---|---|---|---|
| $\ln D_{ijt}$ | | | | | | | | | | | |
| $\ln P_{it}$ | 0.938 (0.016) *** | 0.939 (0.016) *** | 0.938 (0.016) *** | 0.938 (0.016) *** | 0.939 (0.016) *** | 0.939 (0.016) *** | 0.938 (0.016) *** | 0.937 (0.016) *** | 0.937 (0.016) *** | 0.957 (0.016) *** | 0.938 (0.016) *** |
| $\ln P_{jt}$ | 0.988 (0.020) *** | 0.989 (0.020) *** | 0.984 (0.020) *** | 0.985 (0.020) *** | 0.988 (0.020) *** | 0.987 (0.020) *** | 0.987 (0.020) *** | 0.984 (0.020) *** | 0.983 (0.020) *** | 1.041 (0.020) *** | 0.988 (0.020) *** |
| Common language ij | | | | | | | | | | | |
| Colony | | | | | | | | | | | |
| Landlocked i | | | | | | | | | | | |
| Landlocked j | | | | | | | | | | | |
| Common border ij | | | | | | | | | | | |
| BFTA ij | -0.022 (0.062) | -0.583 (0.321) * | 0.283 (0.110) ** | 0.268 (0.084) *** | -0.259 (0.389) | 0.563 (0.270) ** | 0.209 (0.279) | 0.316 (0.120) *** | 0.386 (0.125) *** | -0.781 (0.056) *** | -0.042 (0.047) |
| Observations | 8,060 | 8,060 | 8,060 | 8,060 | 8,060 | 8,060 | 8,060 | 8,060 | 8,060 | 8,060 | 8,060 |
| Exporter FE | YES | YES | YES | YES | YES | YES | YES | YES | YES | YES | YES |
| Importer FE | YES | YES | YES | YES | YES | YES | YES | YES | YES | YES | YES |
| Year FE | YES | YES | YES | YES | YES | YES | YES | YES | YES | YES | YES |

Note: Significant. codes: 0.01 '***' 0.05'**' 0.1 '*'. Robust standard errors are between parentheses.

By applying time and country (exporter and importer) fixed effects, the variables that are invariant to time are omitted.

during the pre- and post-FTA periods. In all post-FTA cases, the averages of U.S. imports have seen a tremendous increase compared with the pre-FTA periods. The highest average among all post-FTAs can be observed in Oman's case, with an increase of 388% in the flow of U.S. imports. Peru showed an increase of 383%, followed by Morocco, at 381%. The lowest increases in the averages of U.S. imports across FTAs are seen in Singapore and the Republic of Korea, with an increases of 174% and 166%, respectively.

As previously mentioned, 8,060 observations were used in the PPML regression. The explanatory variables revealed the country-specific estimates displayed in Table 8, which

**Table 7. Bilateral U.S imports from trade partners before and after FTAs entered into force.**

| FTA partner | Before | After | Total import | | Average import | |
|---|---|---|---|---|---|---|
| | | | Pre-FTA | Post-FTA | Pre-FTA | Post-FTA |
| Australia | 1992–2004 | 2005–2017 | 149,072,702,794 | 299,223,540,589 | 11,467,130,984 | 23,017,195,430 |
| Bahrain | 1992–2005 | 2006–2017 | 5,578,371,761 | 11,343,321,064 | 398,455,125.8 | 945,276,755.3 |
| Chile | 1992–2003 | 2004–2017 | 38,920,840,776 | 166,424,660,066 | 3,243,403,398 | 11,887,475,719 |
| Colombia | 1992–2011 | 2012–2017 | 120,614,629,005 | 97,518,521,197 | 6,030,731,450 | 16,253,086,866 |
| Jordan | 1992–2001 | 2002–2017 | 3,273,100,000 | 18,993,840,929 | 327,310,000 | 1,187,115,058 |
| Morocco | 1992–2005 | 2006–2017 | 6,878,499,458 | 22,500,900,955 | 491,321,389.9 | 1,875,075,080 |
| Oman | 1992–2008 | 2009–2017 | 7,353,360,765 | 15,143,096,576 | 432,550,633.2 | 1,682,566,286 |
| Panama | 1992–2012 | 2013–2017 | 61,157,763,668 | 46,012,640,997 | 2,912,274,460 | 9,202,528,199 |
| Peru | 1992–2008 | 2009–2017 | 36,871,284,394 | 74,852,980,945 | 2,168,899,082 | 8,316,997,883 |
| Singapore | 1992–2003 | 2004–2017 | 184,249,974,923 | 375,885,369,000 | 15,354,164,577 | 26,848,954,929 |
| South Korea | 1992–2011 | 2012–2017 | 526,537,833,547 | 262,677,390,814 | 26,326,891,677 | 43,779,565,136 |

**Table 8. PPML estimation results without fixed effects for the United States as an importer.**

| | AUS-USA | BHR-USA | CHL-USA | COL-USA | JOR-USA | MOR-USA | OMN-USA | PAN-USA | PER-USA | SGN-USA | KOR-USA |
|---|---|---|---|---|---|---|---|---|---|---|---|
| $\ln D_{ijt}$ | -0.741 (0.035) *** | -0.754 (0.035) *** | -0.755 (0.035) *** | -0.762 (0.035) *** | -0.755 (0.035) *** | -0.756 (0.035) *** | -0.753 (0.035) *** | -0.759 (0.035) *** | -0.758 (0.035) *** | -0.781 (0.035) *** | -0.764 (0.035) *** |
| $\ln P_{it}$ | 0.887 (0.007) *** | 0.888 (0.007) *** | 0.888 (0.007) *** | 0.888 (0.007) *** | 0.888 (0.007) *** | 0.887 (0.007) *** | 0.888 (0.007) *** | 0.888 (0.007) *** | 0.888 (0.007) *** | 0.893 (0.007) *** | 0.889 (0.007) *** |
| $\ln P_{jt}$ | 0.987 (0.008) *** | 0.987 (0.008) *** | 0.987 (0.008) *** | 0.988 (0.008) *** | 0.987 (0.008) *** | 0.987 (0.008) *** | 0.987 (0.008) *** | 0.987 (0.008) *** | 0.987 (0.008) *** | 0.989 (0.008) *** | 0.986 (0.008) *** |
| Common language ij | 0.246 (0.019) *** | 0.239 (0.019) *** | 0.239 (0.019) *** | 0.239 (0.019) *** | 0.239 (0.019) *** | 0.238 (0.019) *** | 0.239 (0.019) *** | 0.239 (0.019) *** | 0.238 (0.019) *** | 0.229 (0.019) *** | 0.243 (0.019) *** |
| Colony | 0.134 (0.104) | 0.147 (0.104) | 0.148 (0.105) | 0.150 (0.104) | 0.148 (0.104) * | 0.147 (0.104) | 0.147 (0.104) | 0.149 (0.104) | 0.149 (0.104) | 0.172 (0.105) * | 0.153 (0.104) |
| Landlocked i | -0.962 (0.104) *** | -0.955 (0.104) *** | -0.953 (0.104) *** | -0.955 (0.104) *** | -0.955 (0.104) *** | -0.960 (0.104) *** | -0.956 (0.104) *** | -0.956 (0.104) *** | -0.957 (0.104) *** | -0.937 (0.103) *** | -0.944 (0.104) *** |
| Landlocked j | -1.019 (0.135) *** | -1.017 (0.136) *** | -1.017 (0.136) *** | -1.016 (0.136) *** | -1.017 (0.136) *** | -1.018 (0.135) *** | -1.018 (0.135) *** | -1.017 (0.135) *** | -1.017 (0.136) *** | -1.012 (0.136) *** | -1.017 (0.135) *** |
| Common border ij | 1.018 (0.053) *** | 1.004 (0.053) *** | 1.003 (0.053) *** | 0.993 (0.053) *** | 1.004 (0.053) *** | 1.001 (0.052) *** | 1.006 (0.053) *** | 0.997 (0.053) *** | 0.998 (0.053) *** | 0.975 (0.053) *** | 0.995 (0.053) *** |
| *BFTA ij* | -0.616 (0.192) *** | -0.523 (0.626) | 0.074 (0.194) | -0.287 (0.215) | -0.122 (0.483) | -1.936 (0.654) *** | -0.880 (0.654) | -2.252 (1.285) * | -0.521 (0.283) * | 1.005 (0.138) *** | 0.807 (0.105) *** |
| Observations | 8,060 | 8,060 | 8,060 | 8,060 | 8,060 | 8,060 | 8,060 | 8,060 | 8,060 | 8,060 | 8,060 |
| Exporter FE | NO | NO | NO | NO | NO | NO | NO | NO | NO | NO | NO |
| Importer FE | NO | NO | NO | NO | NO | NO | NO | NO | NO | NO | NO |
| Year FE | NO | NO | NO | NO | NO | NO | NO | NO | NO | NO | NO |

Note: Significant. codes: 0.01 '***' 0.05'**' 0.1 '*'. Robust standard errors are between parentheses.

presents the PPML estimation results for the United States as an importer without applying fixed effects. As expected, the results in Table 8 shown below, display the distance variable exhibited similar statistical significance, with a negative correlation between the volume of imports from BFTAs to the U.S. In terms of the GDPs of the exporting and importing countries, they all have statistical significance and positive coefficients, showing a positive relationship between GDP and the volume of imports. Once again, the common border and common language variables revealed positive coefficients that were statistically significant, confirming the results in Table 6. The colonization variable (*Colony*) does not have a high statistical significance in this case of the U.S. being an importer and, similar to Table 5, the coefficients were mainly insignificant, except for Jordan and Singapore. Both landlocked i and j variables confirmed statistical significance with negative coefficients, which represents the negative correlation between the volume of goods entering the U.S. and the increase in transportation costs stemming from the absence of easily accessible shipping ports. Finally, the dummy variable showed statistical significance for six BFTAs out of the 11 BFTAs, except for Bahrain, Chile, Colombia, Jordan, and Oman.

Unlike in the case of the U.S. as an exporter, Australia''s coefficient is negative (-0.616) and highly significant. This indicates some weaknesses of the US–Australian BFTA and the need to renegotiate the trade agreement from the Australian side. Similarly, the BFTAs with Morocco, Panama, and Peru also showed negative coefficients (-1.936, -2.252, and 0–521, respectively), meaning worse performing FTAs for goods flowing from those countries to the U.S. Singapore (1.005) and the Republic of Korea (0.807) were the best-performing countries with statistically significant coefficients. Additionally, the intrinsic bias in the estimates was managed by controlling for country–time variables, as presented in Table 9. Once again, the application of

**Table 9. PPML estimation results with country and time fixed effects for the United States as an importer.**

|  | AUS-USA | BHR-USA | CHL-USA | COL-USA | JOR-USA | MOR-USA | OMN-USA | PAN-USA | PER-USA | SGN-USA | KOR-USA |
|---|---|---|---|---|---|---|---|---|---|---|---|
| $\ln D_{ijt}$ |  |  |  |  |  |  |  |  |  |  |  |
| $\ln P_{it}$ | 0.938 (0.016) *** | 0.939 (0.016) *** | 0.938 (0.016) *** | 0.938 (0.016) *** | 0.938 (0.016) *** | 0.938 (0.016) *** | 0.939 (0.016) *** | 0.939 (0.016) *** | 0.939 (0.016) *** | 0.984 (0.016) *** | 0.939 (0.016) *** |
| $\ln P_{jt}$ | 0.988 (0.020) *** | 0.988 (0.020) *** | 0.987 (0.020) *** | 0.988 (0.020) *** | 0.987 (0.020) *** | 0.988 (0.020) *** | 0.988 (0.020) *** | 0.988 (0.020) *** | 0.988 (0.020) *** | 1.013 (0.019) *** | 0.990 (0.020) *** |
| Common language ij |  |  |  |  |  |  |  |  |  |  |  |
| Colony |  |  |  |  |  |  |  |  |  |  |  |
| Landlocked i |  |  |  |  |  |  |  |  |  |  |  |
| Landlocked j |  |  |  |  |  |  |  |  |  |  |  |
| Common border ij |  |  |  |  |  |  |  |  |  |  |  |
| BFTA ij | -0.101 (0.095) | -0.381 (0.400) | 0.189 (0.125) | 0.111 (0.078) | 2.173 (1.213) * | 0.181 (0.346) | -0.208 (0.294) | -0.643 (0.450) | -0.076 (0.125) | -1.248 (0.058) *** | 0.108 (0.039) *** |
| Observations | 8,060 | 8,060 | 8,060 | 8,060 | 8,060 | 8,060 | 8,060 | 8,060 | 8,060 | 8,060 | 8,060 |
| Exporter FE | YES | YES | YES | YES | YES | YES | YES | YES | YES | YES | YES |
| Importer FE | YES | YES | YES | YES | YES | YES | YES | YES | YES | YES | YES |
| Year FE | YES | YES | YES | YES | YES | YES | YES | YES | YES | YES | YES |

Note: Significant. codes: 0.01 '***' 0.05'**' 0.1 '*'. Robust standard errors are between parentheses.

time and country (exporter and importer) fixed effects dictates the inclusion of time-variant variables only. The GDP estimates (Table 9) are similar to those in Table 8; all GDPs were highly significant with positive coefficients. However, only three BFTAs were significant—Jordan, Singapore, and Korea. Jordan and the Republic of Korea showed positive coefficients of 2.173 and 0.108, respectively, while Singapore had a negative coefficient -1.248.

## Discussion

Undoubtedly, the main motive to enter an FTA is to stimulate and boost trade in general and exports in particular. However, there are other implicit and explicit reasons for entering trade agreements, either political or economic, which vary from country to country and according to international and national circumstances. The U.S. International Trade Administration describes the economic goals and political objectives of FTAs as follows:""Trade agreements that reduce the barriers to U.S. exports and protect U.S. interests and enhance the rule of law in the FTA partnership"" [38]. On the one hand, NAFTA aims to""strengthen"" the economic ties and the special bond of""friendship"" and""cooperation"" in addition to the creation of expanded trade deals. On the other hand, the BFTAs between the US and Morocco, Bahrain, and Oman did not appear to boost the level of imports from those countries; in fact, the US–Moroccan agreement has only stimulated US exports rather than imports. This shows non-trade political reasons measured directly by the value of trade flows. Initially, the Advisory Committee on Trade Policy and Negotiations (ACTPN) reports on the Moroccan FTA endorsed it for other benefits beyond trade and employment opportunities, which provide a strong base on which to construct additional bilateral or regional agreements [39]. The ACTPN report referred to this agreement with the political argument "to help promote improved and strengthened relations between the U.S. and countries in the Middle East" [40].

In the case of US exports, the U.S. benefits from seven BFTAs, but not from those with Australia, Jordan, Oman, and Korea. However, when the BFTAs are examined from the U.S.

import standpoint, only two countries benefit from the BFTA apart from the United States—Jordan and Korea—with positive relationships to U.S. imports. The least performing BFTA was Singapore, which displayed a negative association with the trade flow as a whole.

To determine whether the BFTAs have benefited the U.S. or not, we need to understand why some trade agreements were made despite the lack of economic feasibility. Although the literature has shown that FTAs have, on average, lowered tariff rates and non-tariff barriers, such reductions can only be seen as necessary starting points for the industry to take advantage of trade deals [16]. There are three main reasons for some FTAs. First, some FTAs were designed through the lens of US foreign policy agreements and intended to strengthen international relationships. The occurrence of wars, especially in the early 2000s, political tensions, or terrorist events interrupted some FTAs and made them economically ineffective and less impactful from the trade perspective. Such external or domestic events or shocks could limit or delay the FTAs from reaching maturity. Nonetheless, the purpose of some agreements was to build the capacity and skills of the local governments and trading entities to take advantage of the lowered barriers and development of institutional preparedness. In general, the rationale behind smaller agreements is that in the absence of a broader economic agreement, some experts say that smaller FTAs accomplish liberalization and the expansion of markets for U.S. goods. Europe has embraced such deals since the 1950s, but the United States signed its first deal in 1985. There are more than 200 such deals worldwide. On the domestic front, an increasing number of FTAs lowered the price of consumer goods in the United States and the costs that U.S. businesses pay for imported materials. After all, increased competition in local markets spurs innovation, increases labor productivity, and creates a better business environment for U.S. investors.

Politically, some BFTAs are believed to be strategically established by the U.S. with the Middle East. For instance, agreements with Morocco, Jordan, Bahrain, and Oman are regarded by some experts as extending the US political position in the region to build the capacity of institutions within a country or region and maintain the power of states.

While the empirical political studies have been limited, most research papers on trade have confirmed that FTAs have reduced tariff rates and non-tariff obstacles. Such reductions can be seen as essential outset points for trade to benefit from [41]. Other investigations have explained that FTAs showed mixed results. Namely, some consider that FTAs had delivered ineffective results in terms of trade growth for the following reasons: several FTAs were conceived as foreign policy agreements, while others were diverted because of external circumstances or shocks. In some cases, one or both partners did not hold the necessary institutional capacity, as the local authorities needed capabilities or skills to take advantage of the lowered barriers. This indicates that the agreements were more like friendships or political agreements than actual economic trade deals. David Hundt advised that it is difficult to conclude the commercial success (trade-oriented outcomes) of FTAs established initially as foreign economic policy arrangements, caused by the political outlook of government authorities [42].

FTAs often promote policy issues arising from general national concerns and WTO integration issues [43]. Political influence is crucial in selecting a trading partners and settling the specifics of agreements during negotiations [44]. Global price volatilities would affect countries massively reliant on exporting natural resources, such as banana republics. The implementation of an FTA would also become difficult for trade partners who suffer from internal instabilities and may not benefit from such agreements while sinking into political conflict or economic hardship. For instance, a nearby country such as Mexico, which encountered a macro-financial crisis in the mid-nineties, went into several years of intense competition with China, which produced consumer goods with even cheaper labor than Mexico, leading to a controversy on the benefits of being a NAFTA member country [45, 46].

This weak institutional capacity may explain why some FTAs do not perform as intended. A well-designed FTA resulting in no benefits could indicate that the FTA partners may be institutionally handicapped either from a structural viewpoint or due to economic limitations restricting the trade benefits from lowered trade barriers. Without building the institutional capacity that enhances the ease of doing trade, promoting adequate infrastructure for regional connectivity, supporting the development of human capital, and enacting policy to protect intellectual property rights, tariff reductions or elimination will only cause insignificant trade benefits [47]. The characteristics of trade partners and institutions that embody them represent a significant part of the cultivation of trade. Further, corruption, state fragility, and the political system itself could signal the possibility of benefiting from FTAs. For instance, democratic systems promote institutional adoption and implement a robust political foundation for international agreements [48]. Institutional capacity can resonate with policy reforms that encourage the deregulation of local businesses that would indirectly support global trade. Moreover, a mature country might have better education systems and appropriate legal structures and implementation and can thus enhance labor market flexibility and boost innovation [27]. By contrast, some governments continue to implement complicated FTA procedures, especially in the rules of origin even after the FTA enactment, which hinders the FTA's ability to perform [49]. In general, political characteristics may encourage more competitive and resilient trade; thus, some developing economies cannot be assumed to function appropriately given the current international trade intensity. For example, Central American economies, such as most members of the CAFTA-DR, are hindered by the quality of their institutions [45]. From a labor perspective, other non-trade benefits impact human capital resources, which may naturally shift to sectors positively affected by the FTA. This move to other industries is mainly driven by the FTA agreement itself, which gives preferential treatment to one industry, making the FTA influential on the labor market and producing more positive trade results in the long run [27]. However, FTAs and trade flows require many years of negotiation. For example, the hysteresis effect of a trade system, which not only depends on its current state and historical status, shows that the history of past trade flows restricts the current trade system and one-time modifications in the course of the trading system could lead to perpetual consequences [13]. Even in well-established trading countries with significant amounts of bilateral trade but political tension before FTAs, the establishment of FTAs did not significantly change trade flows [49]. Therefore, a more extensive bilateral trade alliance at the intergovernmental level might be needed to maximize the benefits of any trade agreement. Eichengreen and Erwin (1996) recommend preferential trade agreements to be associated with former trade flows [13]. Trading with like-minded countries is often a way to limit resistance. Regarding the low barriers and reduced tariffs, Bergstrand et al. (2011) [14] revealed that the initiation of FTAs shows no differences in imports or exports. The results on the US BFTAs in this study thus require the evaluation of these agreements considering the factors mentioned above to determine whether they have added benefits beyond trade and not necessarily something to do with performance.

From a macroeconomic viewpoint, an FTA might shift trade from national producers to international ones, called trade diversion and trade creation [50]. Some scholars also assume that the creation of BFTAs undermines the multilateral trading order and misrepresents attempts by the WTO to consolidate more open, fair, and internationalized laws [51]. Therefore, the creation of more BFTAs without MFTAs is symbolized by the creation of trade fortresses, which do not improve the global trading system, and FTAs may be seen as obstacles to the efforts of international organizations such as the WTO [52]. Nonetheless, BFTAs have always been appealing politically over MFTAs and are much simpler to negotiate or renegotiate.

## Conclusions

Overall, trade negotiators and entities have claimed to be faithful defenders of FTAs. For instance, the U.S. Chamber of Commerce communicated in 2015: "Taken as a group, the United States ran a trade surplus with its FTA partner countries in 2012 and 2013; however, there were deficits in goods trade with its FTAs, which were $80 billion in 2012 and around $70 billion in 2013" [53].

Therefore, this study adds to the literature on the debate on the effect of FTAs on trade flows by examining bilateral FTAs as a whole, an approach that has received little attention from scholars to date. FTAs assist in lowering trade restrictions between constituent countries, which should improve trade flows. Nevertheless, the discussion about the association between BFTAs and the industry level and volume of trade within industry levels remains inconclusive. Additionally, quantifying political influence within trade is a current limitation of this research. This is due to the lack of empirical evidence; nevertheless, some plausible indicators could be captured in future research to include the political perspective based on measurable variables. An example of such an indicator is the political fragility index, which highlights the determinants of BFTA success. Another direction for future research stems from industry-level examination.

FTA countries that open their economies to foreign industries gain access to international market or provide accessibility to a particular country either for economic motives or strengthening political ties and soft power. It is thus worth examining the industry level to precisely capture the most profitable industries within every BFTA. Future research should aim to divide the dataset into pre- and post-crisis periods, for instance, in relation to the US–China trade war or the COVID-19 pandemic.

The current rationale indicates that FTAs will enhance member countries' capacities to trade goods, thus increasing their national incomes. Consequently, in developed economies, BFTAs contribute directly to trade size and, hence, economic growth, which is the cornerstone of such agreements. However, few BFTAs have revealed an immediate need for renegotiation for fair trade and sustainable growth. In other words, trade agreements in such regions may be established mainly to strengthen the political stability within a particular area rather than to create trade.

## Author Contributions

**Conceptualization:** Saleh Saud AlSaif, Tareq Saeed AlShammari.

**Data curation:** Tareg Ghazi Alghabbabsheh.

**Formal analysis:** Ali M. A. Mahmoud.

**Methodology:** Saleh Saud AlSaif.

**Software:** Tareg Ghazi Alghabbabsheh, Ali M. A. Mahmoud.

**Supervision:** Md. Saiful Islam.

**Visualization:** Tareq Saeed AlShammari.

**Writing – original draft:** Saleh Saud AlSaif.

**Writing – review & editing:** Md. Saiful Islam.

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
