## [Decision Letter · Decision Letter 0]

10 Nov 2021

PONE-D-21-29723Assessment of Bilateral Free Trade Agreements (BFTAs): Lessons from the Existing Twelve U.S. BFTAs between 1992 and 2017PLOS ONE

Dear Dr. Md. Saiful Islam,

Thank you for submitting your manuscript to PLOS ONE. After careful consideration, we feel that it has merit but does not fully meet PLOS ONE’s publication criteria as it currently stands. Therefore, we invite you to submit a revised version of the manuscript that addresses the points raised during the review process.

We look forward to receiving your revised manuscript.

Kind regards,

Ricky Chia Chee Jiun

Academic Editor

PLOS ONE

 “There is no funding.”

“There is no competing interest.”

4. We note that you have stated that you will provide repository information for your data at acceptance. Should your manuscript be accepted for publication, we will hold it until you provide the relevant accession numbers or DOIs necessary to access your data. If you wish to make changes to your Data Availability statement, please describe these changes in your cover letter and we will update your Data Availability statement to reflect the information you provide

6. We note that figure 1 in your submission contain map images which may be copyrighted. All PLOS content is published under the Creative Commons Attribution License (CC BY 4.0), which means that the manuscript, images, and Supporting Information files will be freely available online, and any third party is permitted to access, download, copy, distribute, and use these materials in any way, even commercially, with proper attribution. For these reasons, we cannot publish previously copyrighted maps or satellite images created using proprietary data, such as Google software (Google Maps, Street View, and Earth). For more information, see our copyright guidelines: http://journals.plos.org/plosone/s/licenses-and-copyright.

7. We note you have included a table to which you do not refer in the text of your manuscript. Please ensure that you refer to Tables 4 and 5 in your text; if accepted, production will need this reference to link the reader to the Table.

Reviewers' comments:

Reviewer's Responses to Questions

Review Comments to the Author

Reviewer: Assessment of Bilateral Free Trade Agreements (BFTAs): Lessons from the Existing Twelve U.S. BFTAs between 1992 and 2017

Comments

The manuscript describes the issue of the performance of Bilateral Free Trade Agreements (BFTA)between the United States and twelve nations using a multi-dimensional approach. The issue is important and addresses a research question that matters for the impact on the trading behavior of the United States.

However, it could be improved if the impact of the empirical findings is linked better with the research questions and the extant empirical finding in the literature. This article describes the lack of literature review and opinions, especially in the introduction. It is recommended to restructure and update the latest article or use the literature that matches the data.

Some of the most important specific Comments:

Abstract

The author should explain how the role of BFTA in the impact on trade restrictions differs from with existing literature. The abstract should be extended and provide broader insight into the paper.

Introduction

In the introduction, there is little mention opinions of the impact of BFTA from the existing literature.

Methodology and results

It is essential to better describe the methodology for readers. Description of results is less connected with the difference from countries, not clearly corresponding with methodology and variables. I strongly recommend authors adjust the presentation of the table. Also, I suggest that authors can do the robustness test, as well as the industry- and country-level tests, before and after the free trade agreements tests, the multilateral trade resistance effect, and crisis, contagion, and BFTA spillover.

Conclusions, Implications and limitations

The implications of the results carried out, the limitations and the future lines of research are less.

Here are my general comments about the paper:

1. Firms were chosen from twelve countries. However, what specific country that these countries were chosen is not clear. Global manufacturing has changed the way U.S. multinational companies make investment decisions, and FTA partners have a large number of production sharing activities in the manufacturing industry. Therefore, the choice of country is important to understand the proportion of BFTA and how these changes throughout the period studied would be an important observation.

2. We suggest authors use the Figure to show the trends of aggregated imports and exports to BFTA countries from the U.S. by trade flow types.

3. Each country in the observation has been affected by the BFTA at different levels. In my opinion, observing the impact of the Log-Real GDP on these countries may lead to biased results in terms of the effect of preferential tariff reductions and tariff margins. I suggest that the effect of agreements will need to adjust for the endogeneity of BFTA.

4. The author's explanation of the possibility of the result is reasonable. However, it is not easy to find from the empirical results that it contains the variables used. For example, the author mentioned that Political influence and intellectual property rights are both factors that lead to the success of BFTA. Therefore, if there are similar findings empirically, it may highlight the contribution.

5. According to Baier & Bergstrand (2007), the presence of the endogeneity may lead to the debate for stable estimates, that is, it may be biased, and the effects of FTAs on trade may be over-or under-estimated. However, it is not a clear discussion or conjecture of explanations in this study.

6. Robustness test applied to OLS for validity and reliability of the study is not clear. The dataset may be divided into before, and after the BFTA, and the period of crisis such as the US-China trade war or the Covid-19 pandemic; that is, for a view of policy perspective for a regional power within the free trade area.

Best of Luck

Reference

Leung, J. Y. (2016). Bilateral vertical specialization between the US and its trade partners—before and after the free trade agreements. International Review of Economics & Finance, 45, 177-196.

Baier, S. L., & Bergstrand, J. H. (2007). Do free trade agreements actually increase members' international trade?. Journal of international Economics, 71(1), 72-95.

---

## [Author Response · Author response to Decision Letter 0]

25 Jan 2022

Response to Reviewers

We thank the unanimous reviewers for thoughtful suggestions and insights, which have enriched the manuscript and produced a better and more balanced account of the research.

Comments from Reviewer 1:

Comment: The manuscript describes the issue of the performance of Bilateral Free Trade Agreements (BFTA)between the United States and twelve nations using a multi-dimensional approach. The issue is important and addresses a research question that matters for the impact on the trading behavior of the United States. However, it could be improved if the impact of the empirical findings is linked better with the research questions and the extant empirical finding in the literature. This article describes the lack of literature review and opinions, especially in the introduction. It is recommended to restructure and update the latest article or use the literature that matches the data.

Response: These comments have been addressed with a review of the literature and opinions in the introduction having been added to the revised paper. The revised literature review is in line with the provided data.

Comment: the author should explain how the role of BFTA in the impact on trade restrictions differs from with existing literature. The abstract should be extended and provide broader insight into the paper.

Response: The revised abstract describes the debate on the effect of FTAs on trade flows and the application of the pre-FTA and post-FTA analyses to compare and contrast the volumes of exports between these periods.

Comment: in the introduction, there is little mention opinions of the impact of BFTA from the existing literature.

Response: The revised introduction highlights the common debates in FTAs proposed by the hard proponents and opponents of free trade. Opnions of international organizations have also been included to show the trends in multilateral discussions.

Comment: Each country in the observation has been affected by the BFTA at different levels. In my opinion, observing the impact of the Log-Real GDP on these countries may lead to biased results in terms of the effect of preferential tariff reductions and tariff margins. I suggest that the effect of agreements will need to adjust for the endogeneity of BFTA. 

According to Baier & Bergstrand (2007), the presence of the endogeneity may lead to the debate for stable estimates, that is, it may be biased, and the effects of FTAs on trade may be over-or under-estimated. However, it is not a clear discussion or conjecture of explanations in this study.

Response: We have included another diagnostic test. The methodology now includes country and time fixed effects to account for any possible endogeneity.

Comment: Robustness test applied to OLS for validity and reliability of the study is not clear. The dataset may be divided into before, and after the BFTA, and the period of crisis such as the US-China trade war or the Covid-19 pandemic; that is, for a view of policy perspective for a regional power within the free trade area.

Response: The OLS results have been removed from the revised paper. The dataset is now divided into before and after the BFTA, as shown in Tables 4 and 7. The study covers the period from 1992 to 2017. As a result, the US–China trade war or the COVID-19 pandemic are not included.

Comment: it is essential to better describe the methodology for readers. Description of results is less connected with the difference from countries, not clearly corresponding with methodology and variables. I strongly recommend authors adjust the presentation of the table.

Response: The descriptions of the methodology has been enhanced and the presentation of the tables has been adjusted to describe the estimated results as mentioned in the methodology and data. Please see Tables 5, 6, 8, and 9.

Comment: Also, I suggest that authors can do the robustness test, as well as the industry- and country-level tests, before and after the free trade agreements tests, the multilateral trade resistance effect, and crisis, contagion, and BFTA spill over.

Response: The paper now includes a robustness check by using the PPML approach with different techniques, such as including country and time fixed effects and excluding other fixed effects. Furthermore, the study presents the before and after FTA results (Tables 4 and 7). The research focuses on the aggregate data, but because of data limitations, we did not include industry- and country-level tests; multilateral trade resistance effect; and crisis, contagion, and BFTA spill over. Future research could be carried out to investigate these points.

Comment: the implications of the results carried out, the limitations and the future lines of research are less.

Response: All limitations and direction for future research are considered and addressed in the revised manuscript.

Comment: Firms were chosen from twelve countries. However, what specific country that these countries were chosen is not clear. Global manufacturing has changed the way U.S. multinational companies make investment decisions, and FTA partners have a large number of production sharing activities in the manufacturing industry. Therefore, the choice of country is important to understand the proportion of BFTA and how these changes throughout the period studied would be an important observation. We suggest authors use the Figure to show the trends of aggregated imports and exports to BFTA countries from the U.S. by trade flow types.

Response: We revised the number of countries included in the study. Eleven nations instead of 12 have been considered and the established FTAs during the observation period. The historical background section has been extended and additional figures and a table show the trends of trade (exports vs. imports). 

Comment: the author's explanation of the possibility of the result is reasonable. However, it is not easy to find from the empirical results that it contains the variables used. For example, the author mentioned that Political influence and intellectual property rights are both factors that lead to the success of BFTA. Therefore, if there are similar findings empirically, it may highlight the contribution.

Response: We addressed these issues under limitations and direction for future research.

Comment: Reference: 

Leung, J. Y. (2016). Bilateral vertical specialization between the US and its trade partners—before and after the free trade agreements. International Review of Economics & Finance, 45, 177-196.

Baier, S. L., & Bergstrand, J. H. (2007). Do free trade agreements actually increase members' international trade?. Journal of international Economics, 71(1), 72-95.

Response: Cited and added to the reference list. Thank you.

Comments from Reviewer 2:

Comment: Please ensure that your manuscript meets PLOS ONE's style requirements, including those for file naming. The PLOS ONE style templates can be found at

Response: The manuscript has been updated and the formatting guidelines have been followed.

Comment: Thank you for stating the following financial disclosure:

 “There is no funding.”

Response: The data and the study have not been funded.

Comment: Thank you for stating the following in your Competing Interests section: 

“There is no competing interest.”

Response: The authors declare no competing interests.

Comment: We note that you have stated that you will provide repository information for your data at acceptance. Should your manuscript be accepted for publication, we will hold it until you provide the relevant accession numbers or DOIs necessary to access your data. If you wish to make changes to your Data Availability statement, please describe these changes in your cover letter and we will update your Data Availability statement to reflect the information you provide

Response: To be provided upon acceptance.

Comment: In your Data Availability statement, you have not specified where the minimal data set underlying the results described in your manuscript can be found. PLOS defines a study's minimal data set as the underlying data used to reach the conclusions drawn in the manuscript and any additional data required to replicate the reported study findings in their entirety. All PLOS journals require that the minimal data set be made fully available. For more information about our data policy, please see http://journals.plos.org/plosone/s/data-availability. 

Response: Acknowledged and to be provided upon acceptance.

Comment: We note that figure 1 in your submission contain map images which may be copyrighted. All PLOS content is published under the Creative Commons Attribution License (CC BY 4.0), which means that the manuscript, images, and Supporting Information files will be freely available online, and any third party is permitted to access, download, copy, distribute, and use these materials in any way, even commercially, with proper attribution. For these reasons, we cannot publish previously copyrighted maps or satellite images created using proprietary data, such as Google software (Google Maps, Street View, and Earth). For more information, see our copyright guidelines: http://journals.plos.org/plosone/s/licenses-and-copyright. 

Maps at the CIA (public domain): https://www.cia.gov/library/publications/the-world-factbook/index.htmland
https://www.cia.gov/library/publications/cia-maps-publications/index.html

Response: Acknowledged, Figure 1 has been omitted as a result.

Comment: We note you have included a table to which you do not refer in the text of your manuscript. Please ensure that you refer to Tables 4 and 5 in your text; if accepted, production will need this reference to link the reader to the Table.

Response: The revised manuscript has addressed all mentioned issues; therefore, Tables 4 and 5 have been omitted and new tables have been inserted into the text as per your comment.

---

## [Editor Report · Decision Letter 1]

16 Feb 2022

Have Bilateral Free Trade Agreements (BFTAs) Been Beneficial? Lessons Learned from 11 U.S. BFTAs between 1992 and 2017

PONE-D-21-29723R1

Dear Dr. Md. Saiful Islam,

We’re pleased to inform you that your manuscript has been judged scientifically suitable for publication and will be formally accepted for publication once it meets all outstanding technical requirements.

Kind regards,

Ricky Chia Chee Jiun

Academic Editor

PLOS ONE
---

## [Editor Report · Acceptance letter]

30 Mar 2022

PONE-D-21-29723R1 

Have Bilateral Free Trade Agreements (BFTAs) Been Beneficial?
Lessons Learned from 11 U.S. BFTAs between 1992 and 2017 

Dear Dr. Islam:

I'm pleased to inform you that your manuscript has been deemed suitable for publication in PLOS ONE. Congratulations! Your manuscript is now with our production department. 

Kind regards, 

on behalf of

Dr. Ricky Chee Jiun Chia 

Academic Editor

PLOS ONE